# Evaluation of Automatic Signal Detection of In Situ Hybridization for Detecting HPV DNA in Cervical Tissue Derived from Patients with Cervical Intraepithelial Neoplasia

**DOI:** 10.3390/cancers16203485

**Published:** 2024-10-15

**Authors:** Marcin Przybylski, Sonja Millert-Kalińska, Mateusz de Mezer, Monika Krzyżaniak, Paweł Kurzawa, Jakub Żurawski, Robert Jach, Dominik Pruski

**Affiliations:** 1Department of Obstetrics and Gynaecology, District Public Hospital, 60-479 Poznan, Poland; nicramp@poczta.onet.pl (M.P.); millertsonja@gmail.com (S.M.-K.); 2Doctoral School, Poznan University of Medical Sciences, 60-812 Poznan, Poland; 3Department of Immunobiology, Poznan University of Medical Sciences, 60-812 Poznan, Poland; mdemezer@ump.edu.pl (M.d.M.); zurawski@ump.edu.pl (J.Ż.); 4Department of Pathomorphology, University Clinical Hospital in Poznan, 60-572 Poznan, Poland; monika.krzyzaniak@skpp.edu.pl (M.K.); paul.kurzawa@yahoo.com (P.K.); 5Department of Gynecological Endocrinology, Jagiellonian University Medical College, 31-008 Kraków, Poland; jach@cm-uj.krakow.pl; 6Gynecology Specialised Practise, 60-408 Poznan, Poland

**Keywords:** automatic signal detection, automatic response field, FFPE, ISH, HPV DNA, CIN, SIL, in situ Hybridization

## Abstract

**Simple Summary:**

Cervical cancer is the fourth most common cancer in women worldwide, with HPV being a prevalent cause. Recent automated signal detection methods in ISH assays have shown promise for detecting HPV DNA in cervical tissue. This study compared an ISH probe (Inform HPV II and III) to PCR assays in cervical tissue samples with varying degrees of cervical intraepithelial neoplasia (CIN) and normal cervix tissue. Findings indicated a significant association between ISH III levels and HPV outcomes; patients with positive outcomes had lower ISH III levels (MD = −7961.82, *p* = 0.005). However, automatic signal detection for ISH is limited in cervical tissue, making genotyping-based HPV testing more effective, as it allows for larger sample collection. ISH results should be interpreted alongside clinical history and morphology.

**Abstract:**

**Background:** Cervical cancer is fourth the most common cancer in women worldwide. Due to the prevalence of human papillomavirus (HPV) in the population (80–90%), scientists are likely to discover even more associations of this pathogen with other diseases in the future. In recent years, In Situ Hybridization (ISH) assays that use automated signal-detecting methods in formalin-fixed, paraffin-embedded (FFPE) cervical tissue, such as the enzyme-categorized signal-detecting system, have shown a higher sensitivity. **Objectives and Methods:** To evaluate automatic signal detection of ISH assay for detecting HPV DNA, we compared the ability of an ISH probe, Inform HPV II and III (Ventana Medical Systems, Tucson, AZ), to that of PCR assays to detect HPV DNA in cervical tissue specimens with cervical intraepithelial neoplasia (CIN; CIN 1, 28 cases; CIN 2, 22 cases; and CIN 3, 20 cases) and normal cervix (2 cases). **Results:** Our findings showed a significant relation was confirmed between ISH III level and HPV outcome (positive/negative). Patients with positive HPV outcomes had significantly lower ISH III levels, MD = −7961.82 CI_95_ [−17,230.00; −199.21], *p* = 0.005. **Conclusions:** Automatic signal detection of ISH assay is not particularly applicable to cervical tissue material. A more useful method of confirming the presence of HPV in the cervix is the HPV test with genotyping, as it allows for collecting a larger amount of material from the cervical disc and canal. The interpretation of a positive or negative ISH test must be guided in the context of clinical history and morphology.

## 1. Introduction

Cervical cancer is the fourth most common cancer in women worldwide GLOBOCAN 2020 estimated that there were approximately 604,000 new cases of cervical cancer, with 342,000 deaths annually [1]. Each year in the USA, 4000 women are diagnosed with cervical cancer and approximately 100,000 women are treated for cervical precancerous lesions [2]. Persistent infection with highly oncogenic HPV types is responsible for developing precancerous lesions and cervical cancer. Additionally, HPV is responsible for forming many other HPV-related changes, including head and neck, vagina, vulva, penis, and anal cancers.

HPV is an approximately 7900 base pair double-stranded DNA virus with over 200 genotypes [3,4,5]. The life cycle of HPV is directly related to the differentiation of keratinocytes. The critical event in the virus life cycle is the escalation of its replication associated with this differentiation. Due to the prevalence of HPV in the population (80–90%), scientists are likely to discover even more associations of this pathogen with other diseases in the future. That is why it is so important to continue searching for diagnostic methods and validate the existing ones and use the best-known and the highest sensitivity and specificity methods for patients. Recent studies compared conventional cytodiagnostics with molecular identification of HPV HR DNA and mRNA, immunocytochemical testing for the suppressor protein P16, and nuclear Ki 67 to detect cervical pathology. However, the superiority of molecular methods over classical methods is becoming more and more common [6,7,8].

In histopathology, HPV DNA testing in formalin-fixed, paraffin-embedded (FFPE) cervical tissue serves several essential purposes. Firstly, they might resolve diagnostic discrepancies in cervical intraepithelial neoplasia (CIN) patients. It aids in distinguishing between endocervical immature squamous metaplasia and high-grade dysplasia, as well as differentiating endocervical glandular reactive changes from glandular dysplasia. Secondly, risk assessment; the test provides valuable information for assessing the risk of CIN progression or disease recurrence in women who have undergone treatment for HSIL (CIN 2/3) or carcinoma. Ultimately, HPV DNA testing contributes crucial data in cervical cancer research, facilitating an understanding of the virus’s role in disease development and progression [9,10].

By employing HPV DNA testing in FFPE cervical tissue, medical professionals may enhance diagnostic accuracy, provide more personalized patient care, and contribute to advancing cervical cancer research. However, there are disadvantages to consider, that the assay requires highly trained laboratory personnel, and strict laboratory conditions must be implemented to avoid contamination [11,12]. The advantages connected with ISH over detecting HPV in tissue by PCR include the lack of destroying the tissue, correlation with cytology results, the ability to test archival samples, and identification of the HPV ISH result in the context of tissue morphology.

In recent years, ISH assays that use improved signal-detecting methods, such as the enzyme-categorized signal-detecting system, have shown a higher sensitivity [13]. Inform HPV (Ventana Medical Systems), a commercially available ISH assay for HPV DNA testing, can be used in cytological and histological specimens [14]. Recently, Inform HPV III, a new generation of ISH probe, became available for HPV DNA testing in tissue specimens. Knowing the efficacy of the ISH assay using the Inform HPV III probe in cervical tissue not only may allow for better use of ISH-based HPV DNA testing in tissue specimens, but also could provide valuable information for the cytological application of ISH in LBC.

Therefore, in the present study, we aim to assess the association between HPV infection and an automatic signal detection of ISH for detecting HPV DNA using the Inform HPV II and HPV III probe in FFPE cervical tissue specimens with CINs.

## 2. Materials and Methods

### 2.1. Study Design

We provide a prospective, ongoing 12-month, non-randomized study in patients reporting to the Individual Specialized Medical Practice and District Public Hospital in Poznan, Poland in 2022–2023. Subjects attended the medical practice as part of in-depth diagnostics due to an abnormal cytological result or the presence of HPV 16, 18, and 31 in the cervical smear and histopathologically confirmed cervical intraepithelial neoplasia. The histopathologist re-verified the diagnosis and marked the places of CIN lesions on the formalin-fixed, paraffin-embedded (FFPE) cervical tissues. The laboratory diagnostician prepared new slide preparations for ISH staining. In two cases, the final medical consensus did not confirm the diagnosis of CIN; therefore, they were excluded from further analysis. The Poznan University of Medical Sciences Bioethical Committee approved the study protocol (540/22) on 23 June 2022.

### 2.2. HPV Genotyping Test and LBC

We collected liquid-based cytology and molecular assessment samples with an endocervical Cyto-Brush preserved in BD SurePath^®^ (Becton, Dickinson and Company, Franklin Lakes, NY, USA). Then, the probes were passed to an independent, standardized laboratory. PCR was performed, followed by a DNA enzyme immunoassay and genotyping with a reverse hybridization line probe assay for HPV detection. The lab technicians performed sequence analysis to characterize HPV-positive samples. The molecular test detected the DNA of 37 HPV genotypes (6, 11, 16, 18, 26, 31, 33, 35, 39, 40, 42, 45, 51, 52, 53, 54, 55, 56, 58, 59, 61, 62, 64, 66, 67, 68, 69, 70, 71, 72, 73, 81, 82, 83, 84, IS39, and CP6108).

### 2.3. Colposcopy and Punch Biopsy

Further validation of abnormal screening results was performed on all patients with an abnormal smear: ASC-US, LSIL, HSIL, ASC-H, AGC, cervical cancer, a positive HPV test for types 16, 18, and 31, and a clinically suspicious cervical image. Each colposcopy was performed by a specialist in gynecologic oncology with 10 years of experience in SmartOPTIC colposcope. We performed a test with a 5% aqueous solution of acetic acid and Schiller’s test with Lugol’s iodine in all included cases. The colposcopic images were evaluated according to Reid’s Colposcopic Index, which assesses the color, lesion boundaries, and surface, blood vessels, and results of the iodine test. All colposcopic images were archived. The Polish Society of Colposcopy and Cervical Pathophysiology recommended the International Federation of Cervical Pathology and Colposcopy classification [15,16,17].

### 2.4. Immunohistochemistry

Serial 4-micrometer tissue sections were cut from the donor blocks containing cores of lesions and applied to special immunohistochemistry-coated slides. Two ISH probes—INFORM HPV II Family 6 Probe and INFORM HPV III Family 16 Probe (Ventana, Roche, Tucson, AZ, USA)—were used to target the common HPV genotypes in cervical biopsy specimens. To demonstrate positive hybridization to low-risk genotypes 6 and 11, the INFORM HPV II Family 6 Probe and the INFORM HPV III Family 16 Probe were used to demonstrate positive hybridization to the following genotypes: 16, 18, 31, 33, 35, 45, 52, 56, 58, and 66.

Slides were stained on a fully automated immunohistochemistry slide stainer, BenchMark ULTRA (Ventana, Roche, Tucson, AZ, USA). Staining protocol parameters were based on HIER using CC2 (heating time 4 + 8 + 8 min at 86 °C), ISH-Protease 3 (780-4149) for 16 min, 12 min of denaturation, and 2 h of hybridization with each ISH probe. To detect specific DNP-labeled probes bound to a target sequence, an indirect biotin-streptavidin system, INFORM iView Blue + (760-097), was used. Slides were then post-counterstained with Red Stain II (780-2218) (Ventana, Roche, Tucson, AZ, USA) for 4 min. Coverslips were passed through a series of alcohols and finally xylene before being mounted.

### 2.5. Light Microscopy Techniques for Cell Imaging

Photographs of the tissue slides were taken, using an Olympus BX 43 light microscope with an XC 30 digital camera (Olympus, Tokyo, Japan). Magnification was set at 400×. Based on the obtained images from the light microscope, a semi-quantitative analysis of immunopositive cells was performed. Calculations were made using the Olympus cellSens commercial software. Phase analysis of immunohistochemically stained tissue microarrays was undertaken, including automatic signal detection of objects based on their color, shade intensity, or shape. In this case, the color criterion—the blue of the cell nuclei due to the ISH reaction—was selected. The computer program automatically classified cells based on predefined threshold values. The data were exported to MS Excel files and used for further statistical analyses [18,19,20]. Figure 1 presents a detection of HPV DNA (in situ hybridization: a—ISH HPV II and ISH b—HPV III) in epithelial cells in the tissue of four patients.

### 2.6. Statistical Analysis

Analysis was conducted with statistical software R, version R4.1.2. All calculations assumed a significance level of α = 0.05. Nominal variables were presented as n and %, while quantitative ones were presented as median with quartiles 1 and 3. The normality of the variable distributions was analyzed with Shapiro–Wilk’s test and further verified with skewness and kurtosis. All tests that were used in the analysis were non-parametric—to compare the level of ISH II and the level of ISH III between groups, Mann–Whitney’s U test, or Kruskal–Wallis test were conducted (depending on the number of groups). Spearman’s correlation analysis was used to analyze the dependency between two quantitative variables. The Kappa coefficient was used to determine the agreement between CIN1, CIN2, and CIN3 with HPV outcomes (positive/negative). We assessed the sensitivity, specificity, PPV (positive predictive value), NPV (negative predictive value), and accuracy with a 95% confidence interval of diagnostic abilities of LBC result, ISH, and HPV outcomes for CIN2 + CIN3, as well as for ISH II for HPV 6 or 11 and ISH III for HPV 16, 18, 31, 33, 35, 45, 52, 56, 58, and 66.

## 3. Results

The sample included data from seventy-two patients aged from 18 to 79 years, with a mean age of 33. The characteristics of the study group are shown in Table 1. Figure 1 shows a detection of HPV DNA in epithelial cells in the tissue of four patients with histopathologically confirmed LSIL. The tissue from patient 1 was determined as ASC-US in LBC while the genotype of the HPV was negative. In turn, the tissue of patient 2 belongs to ASC-US, while the genotype of the HPV was positive for 51. The tissue from patient 3 was classified as ASC-US in LBC and was positive for HPV 16 and 56. Finally, tissue from patient 4 was classified as ASC-H and was positive for HPV 82.

Table 2 shows the relationships of ISH II levels to various variables, but no statistically significant associations were observed. A significant relation was confirmed between ISH III level and HPV outcome (positive/negative). Patients with positive HPV outcomes had significantly lower ISH III levels, MD = −7961.82 CI_95_ [−172,30.00; −199.21], *p* = 0.005. ISH III levels split into HPV groups were visualized in Appendix A. There was a significant difference in ISH III levels between groups with HPV genotypes of 6 and/or 11 and other patients. Patients with HPV genotypes 6 and/or 11 had significantly lower ISH III levels than others, MD = −317.05 CI_95_ [−1972.19; −13.71], *p* = 0.037. ISH III levels in the split between the HPV 6 and/or 11 group and other patients were visualized as presented in Appendix A. No other relations with ISH III were found significant.

No correlation was detected between age and ISH II level: rho = 0.00 CI95 [−0.23; 0.24]; *p* = 0.976, as presented in Figure 2.

A positive correlation of minor strength was detected between the age and ISH III level: rho = 0.23 CI95 [0.00; 0.44]; *p* = 0.049, as shown in Figure 3. With increasing age, there are higher levels of ISH III; however, the strength of this relationship is not high.

No correlation was detected between ISH II and ISH III levels: rho = 0.04 CI95 [−0.20; 0.27]; *p* = 0.760, as presented in Figure 4.

Table 3 compares the value and usefulness of various diagnostic methods in detecting CIN 2+ lesions. In the case of cytological smears, the specificity of CIN 2+ detection increased from the diagnosis of LSIL (46.67%), through ASC-H (86.67%), to the diagnosis of HSIL (96.67%). However, in the case of ASC-US diagnosis with low sensitivity, a relatively high specificity was achieved—76.67%. This result supports the unquestionable necessity of extending diagnostics with HPV DNA molecular testing and cervical biopsy. The sensitivity of HPV testing in detecting CIN 2+ lesions reached the level of 95.24%—higher than any cytological result—which proves the superiority of this test over LBC. Kappa coefficients were calculated to understand the level of agreement between CIN1, CIN2, and CIN3 outcomes with HPV outcome (positive/negative), as presented in Table 4. We are unable to assess the sensitivity of ISH since only histopathological materials with a positive reaction to both ISH II and ISH III were collected for the examination, as shown in Table 5.

## 4. Discussion

### 4.1. Summary of Main Findings

Our study was intended to assess the usefulness of the automatic signal detection of ISH assay using the Inform HPV II and HPV III in diagnosing cervical intraepithelial lesions in formalin-fixed, paraffin-embedded tissues. Additionally, we aimed to describe the sensitivity and specificity of LBC, and HPV DNA testing in relation to automatic signal detection of ISH assay. The main results indicate a significant association between ISH III level and HPV outcome in a study group. Patients with positive HPV results from cervical swabs had significantly lower ISH III levels. This observation may be explained by the fact that the precancerous lesions of the cervix were caused by transient HPV infection or the low level of HPV ISH resulting from the lack of ongoing virus replication. Furthermore, low ISH III levels with a positive HPV test result indicate a much higher HPV DNA value of the cervical smear as it provides a large amount of relevant material. Positive results of automatic signal detection of ISH may also be related to artifacts on the slide preparation, lack of an appropriate amount of tissue material in the preparation, or contamination related to the preparation of the preparation—coming from the person preparing it. However, it seems to us that the assessment of HPV in the tissue by automatic signal detection of ISH may serve other potentially HPV-related cancers, e.g., head and neck cancers or oropharyngeal squamous cell carcinoma (OPSCC). Currently, there is no clear consensus on the gold standard for HPV testing in OPSCC.

### 4.2. Generalizability of Results

Multimodality testing could help to reliably identify patients with transcriptionally active high-risk HPV-positive. Due to this reason, researchers from Naples, Italy conducted a study of HPV RNA ISH and p16 IHC on the same slide to detect simultaneously HPV E6/E7 transcripts and p16INK4a overexpression. They revealed that the multiplex HPV RNA ISH /p16 IHC results in the series of both cervical cancers and the oral-oropharyngeal cancers were fully concordant with the previous results achieved through the classic p16 IHC and HPV RNA scope carried out on two different slides [21].

Sheng et al. conducted a study where ISH used a probe targeting multiple hrHPV subtypes, while PCR, genotyping, and p16 IHC were performed on specimens from 27 cases of AIS and CA. A total of 63% of the AIS and CA specimens were HPV-ISH positive in this study. These results confirm that HPV infection occurs in cervical glandular neoplasia with relatively high frequency. HPV DNA was detected in 67% of cases by PCR. Furthermore, 74% of cases were positive for HPV, following HPV genotyping analysis. Combining the results of HPV–ISH and HPV–PCR/genotyping, 22 AIS and CA cases (81.5%) were considered HPV-positive. When two cases of clear cell adenocarcinoma, in which HPV is generally regarded as negative, HPV was detected in 88% of cases of cervical glandular neoplasia [22]. In our study, all samples were also ISH-positive, but HPV was negative in 8/72 patients (10.8%). Additionally, Tase et al. detected HPV DNA in 42.5% of FFPE sections from CA cases using ISH with mixed probes [23]. However, using RNR-RNA ISH, Milde-Langosch et al. found HPV-16 and -18 E6/E7 oncogene expressions in 62.3% of FFPE sections from adenocarcinoma cases [24].

ISH assays can be used in tissue material and thin-layer cytological swabs. A study conducted by Samama B. et al. on thin-layered Pap smears showed that the rate of HPV detection increased with the severity of cytology results. They revealed DNA HPV 14% in minor cellular changes, 55% in AS-CUS, 95% in LSIL, and up to 100% in HSIL and carcinoma. High-risk HPV types were mainly present, alone or associated with low-risk HPV types, whatever the cytological findings. The researchers found a good correlation between ISH and the Hybrid Capture II test results with cervical smears. By using the HC test as the reference method, the sensitivity of the ISH protocol was 87.5% and the specificity 96% [25].

In the vaccination against HPV era and the era of changes in sexual habits, we are looking for new diagnostic methods to facilitate the identification of patients with real progression of HSIL lesions toward cancer, regardless of the number of obstetric histories. In the diagnosis of SIL changes, the identification of E6 E7 transcripts based on mRNA tests is already used [26,27], and population-based preventive vaccinations against HPV in each age group may influence the occurrence of specific types of HPV and reduce the number of persistent virus infections [28,29,30,31].

Additionally, researchers are looking for new diagnostic methods in patients with persistent HR HPV infection after preventive vaccinations, who may have a higher risk of developing epithelial changes toward HSIL and cervical cancer. Molecular methods, such as mRNA, methylation, or ISH, may be used in laryngology or in men—in swabs from under the foreskin of the penile. This is associated with a significantly lower concentration of HPV particles in secretions [15,32].

To conclude, the interpretation of a positive or negative ISH test must be guided in the context of clinical history and morphology and supplemented by appropriate control slide studies and other diagnostic tests. Responsibility for the use of probes, reagents, and methods for preparing stained slides rests with the collaboration of the histopathologist and clinician.

### 4.3. Limitations of the Study

The limitation of the study is that we cannot quite correlate our results to some findings as we did not assess the number of virus copies in the tissue. Additionally, it is a one-center study, which needs to include more patients with longer follow-ups.

## 5. Conclusions

Automatic signal detection of ISH assay is not particularly applicable to cervical tissue material. A more useful method of confirming the presence of HPV in the cervix is the HPV test with genotyping, as it allows for collecting a larger amount of material from the cervical disc and canal. The interpretation of a positive or negative ISH test must be guided in the context of clinical history and morphology. It should be treated with due care and interpreted by the clinician in consultation with the histopathologist. The future use of automated algorithms based on AI to facilitate the identification of morphological changes in epithelial cells of the HSIL or LSIL type, along with the simultaneous intensity of staining will influence the parameters of the diagnosis of pre-cancerous changes in the cervix.

## Figures and Tables

**Figure 1 cancers-16-03485-f001:**
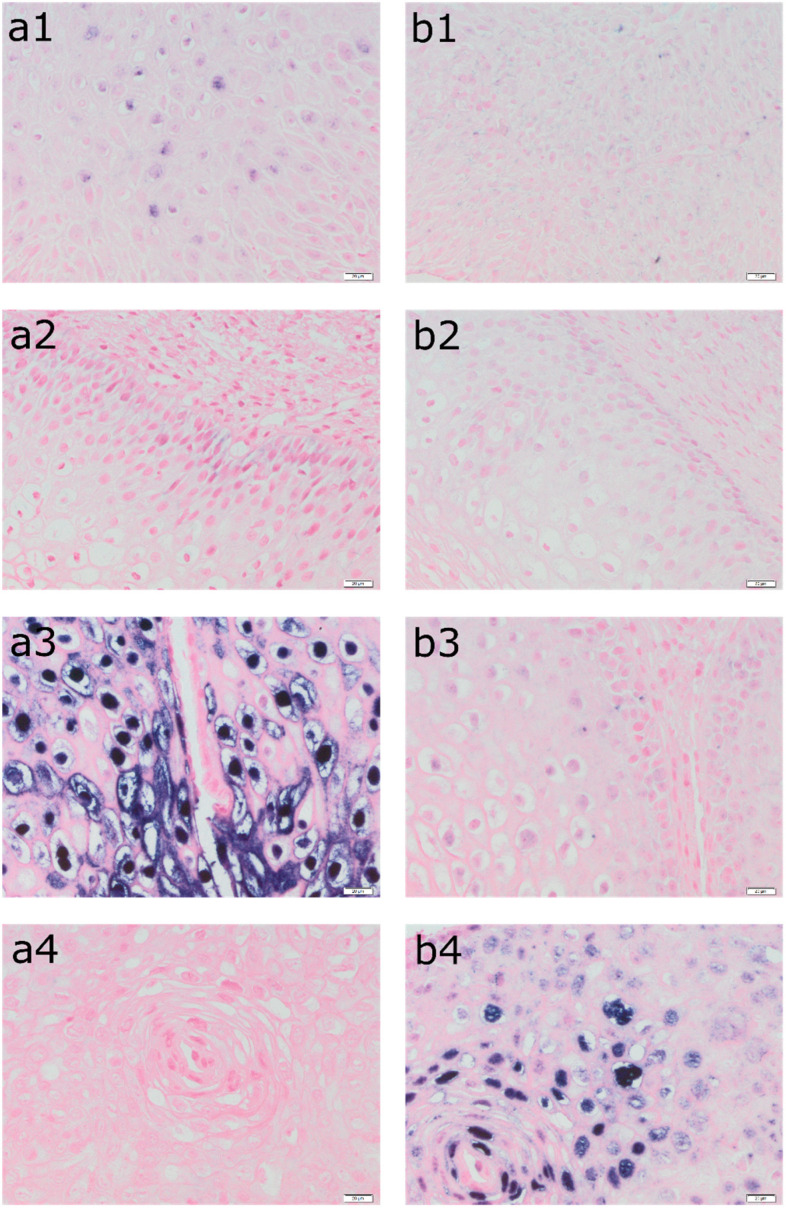
Detection of HPV DNA (in situ hybridization: (**a1**–**a4**)—ISH HPV II and (**b1**–**b4**)—ISH HPV III) in epithelial cells in the tissue of the same patients (1—LBC result: ASC-US, HPV negative, LSIL in biopsy; 2—LBC result: ASC-US, HPV 51 positive, LSIL in biopsy; 3—LBC result: ASC-US, HPV 16, 56 positive, LSIL in biopsy; 4—LBC result: ASC-H, HPV 82 positive, LSIL in biopsy). Magnification 400×. ISH analyses were performed using the INFORM HPV III Family 16 Probe for high-risk HPV (HPV 16, 18, 31, 33, 35, 39, 45, 51, 52, 56, 58, and 66) and the INFORM HPV II Family 6 probe for low-risk HPV (HPV 6 and 11) with the iVIEW Blue Plus Detection Kit Ventana detection system.

**Figure 2 cancers-16-03485-f002:**
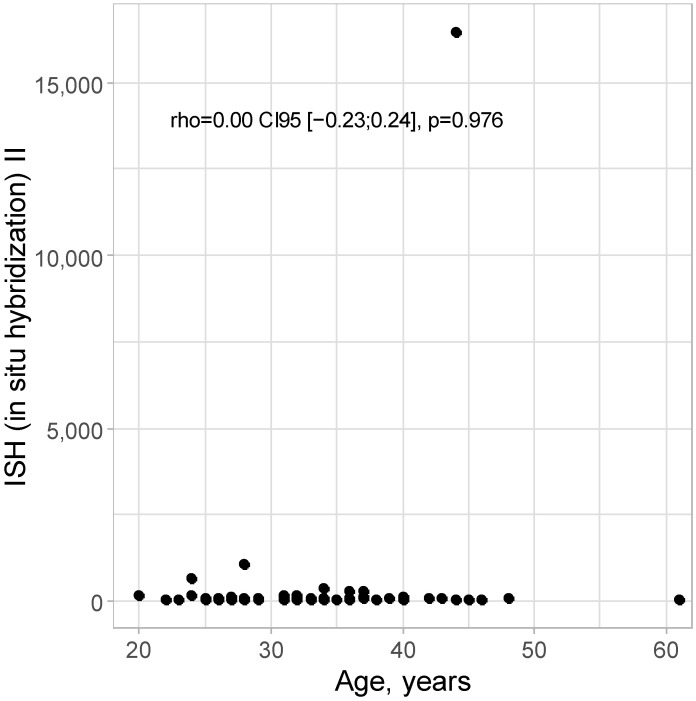
Scatter plot for age and ISH II level.

**Figure 3 cancers-16-03485-f003:**
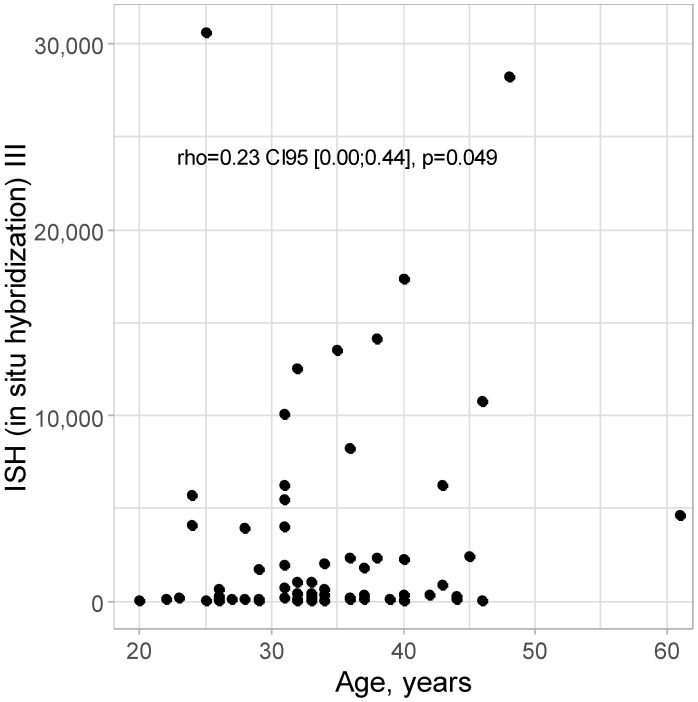
Scatter plot for age and ISH III level.

**Figure 4 cancers-16-03485-f004:**
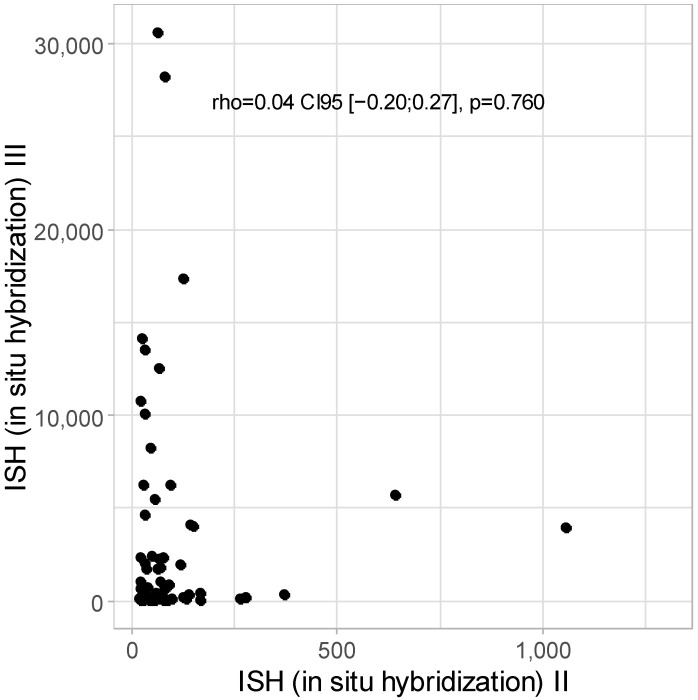
Scatter plot for ISH II and ISH III levels (w/o one patient with ISH II = 16,483.55).

**Table 1 cancers-16-03485-t001:** Group characteristics.

Characteristics	Values
N	72
Age, years, Me (Q1; Q3)	33.00 (28.75; 37.00)
Histopathological diagnosis, n (%)	
CIN1	28 (38.9)
CIN2	22 (30.6)
CIN3	20 (27.8)
No pathology	2 (2.8)
ISH II, Me (Q1; Q3)	58.45 (33.36; 87.99)
ISH III, Me (Q1; Q3)	375.60 (134.90; 2400.16)
LBC result, n (%)	
NILM	3 (4.2)
AS-CUS	11 (15.3)
LSIL	32 (44.4)
ASC-H	15 (20.8)
HSIL	11 (15.3)
HPV, n (%)	
Positive	64 (88.9)
Negative	8 (11.1)
HPV genotype, n (%) *	
6 or 11	8 (11.1)
Any of 16, 18, 31, 33, 35, 45, 52, 56, 58, 66	53 (73.6)
Other positive	22 (30.6)

CIN—cervical intraepithelial neoplasia; LBC—liquid-based cytology; NILM—negative for intraepithelial neoplasia or malignancy; ASC-US—atypical squamous cells of undetermined significance; LSIL—low-grade squamous intraepithelial lesion; ASC-H—atypical squamous cells cannot exclude high-grade squamous intraepithelial neoplasia; HSIL—high-grade squamous intraepithelial lesion; HPV—human papillomavirus; ISH—in situ hybridization; n- number; Me—median, Q1—first quartile, Q3—third quartile. * sum exceeded 100% as patients could belong to multiple groups.

**Table 2 cancers-16-03485-t002:** Comparison of ISH II and ISH III levels between selected groups.

Variables	ISH II Level	MD (95% CI)	*p*	ISH III Level	MD (95% CI)	*p*
LBC result						
NILM	79.12 (52.51; 177.78)	-	-	198.92 (127.20; 220.16)		
AS-CUS	63.55 (44.55; 79.13)	-	0.786	166.58 (147.58; 2379.48)		0.828
LSIL	61.15 (35.89; 92.27)			334.40 (122.33; 2045.52)		
ASC-H	50.77 (32.00; 102.45)			396.79 (215.43; 3932.40)		
HSIL	46.81 (29.84; 69.30)			1031.62 (135.07; 6098.14)		
HPV						
Positive	52.28 (33.12; 85.47)	−28.24 (−88.01; 3.05)	0.072	318.25 (117.75; 1954.46)	−7961.82 (−17,230.00; −199.21)	0.005
Negative	80.52 (67.09; 163.84)			8280.07 (1805.37; 20,100.63)		
HPV genotypes						
6 or 11						
Positive	48.76 (42.74; 54.36)	−14.66 (−34.26; 17.44)	0.809	110.96 (85.23; 195.17)	−317.05 (−1972.19; −13.71)	0.037
Negative	63.42 (32.46; 92.27)			428.01 (153.80; 2828.15)		
Any of 16, 18, 31, 33, 35, 39, 45, 51, 52, 56, 58, 66						
Positive	57.15 (31.72; 86.34)	−8.67 (−41.39; 7.72)	0.188	346.67 (140.08; 1997.89)	−1660.38 (−4089.04; 65.29)	0.157
Negative	65.82 (47.26; 115.24)			2007.05 (141.99; 12,071.24)		
Histopathological diagnosis						
CIN1	63.42 (36.07; 85.47)	11.14 (−16.15; −26.32)	0.527	375.60 (137.87; 3007.58)	−338.61 (−929.97; 516.39)	0.812
CIN2 + CIN3	52.28 (35.59; 71.45)			714.21 (99.78; 2757.10)		

CIN—cervical intraepithelial neoplasia; LBC—liquid-based cytology; NILM—negative for intraepithelial neoplasia or malignancy; ASC-US—atypical squamous cells of undetermined significance; LSIL—low-grade squamous intraepithelial lesion; ASC-H—atypical squamous cells cannot exclude high-grade squamous intraepithelial lesion; HSIL—high-grade squamous intraepithelial lesion; HPV—human papillomavirus; ISH—in situ hybridization; n—number; MD (95% CI)—a median difference with 95% confidence intervals. Comparisons were made with Mann–Whitney’s U when comparing the level of a variable between two groups or with the Kruskal–Wallis test when there were more than two groups. The NILM group was excluded from the Kruskal–Wallis test due to a low number of observations (n = 3).

**Table 3 cancers-16-03485-t003:** Sensitivity and specificity of selected outcomes in CIN2 + CIN3.

Variable		CIN2 + CIN3			Sensitivity, %	Specificity, %	PPV, %	NPV, %	Accuracy, %
		Yes, n = 42	No, n = 30	Total, n = 72					
ISH II	+	42	30	72	100.00 (91.59–100.00)	0.00 (0.00–11.57)	58.33 (56.21–61.17)	-	58.33 (46.95–69.72)
	−	0	0	0					
ISH III	+	42	30	72	100.00 (91.59–100.00)	0.00 (0.00–11.57)	58.33 (56.21–61.17)	-	58.33 (46.95–69.72)
	−	0	0	0					
HPV	+	40	24	64	95.24 (83.84–99.42)	20.00 (7.71–38.57)	62.50 (57.92–66.87)	75.00 (39.38–93.27)	63.89 (52.79–74.98)
	−	2	6	8					
HPV 6 or 11	+	4	4	8	9.52 (2.66–22.62)	86.67 (69.28–96.24)	50.00 (21.34–78.66)	40.62 (36.57–44.81)	41.67 (30.28–53.05)
	−	38	26	64					
HPV group 2 *	+	35	18	53	83.33 (68.64–93.03)	40.00 (22.66–59.40)	66.04 (58.49–72.85)	63.16 (43.37–79.33)	65.28 (54.28–76.27)
	−	7	12	19					

CIN—cervical intraepithelial neoplasia; LBC—liquid-based cytology; NILM—negative for intraepithelial neoplasia or malignancy; ASC-US—atypical squamous cells of undetermined significance; LSIL—low-grade squamous intraepithelial lesion; ASC-H—atypical squamous cells cannot exclude high-grade squamous intraepithelial lesion; HPV—human papillomavirus; n—number; PPV—positive predictive value; NPV—negative predictive value. Comparisons were made with Mann–Whitney’s U when comparing the level of a variable between two groups or with the Kruskal–Wallis test when there were more than two groups. The NILM group was excluded from the Kruskal–Wallis test due to a low number of observations (n = 3). Group 2 *—HPV 16, 18, 31, 33, 35, 45, 52, 56, 58, 66.

**Table 4 cancers-16-03485-t004:** Kappa coefficients for assessing agreement between CIN and HPV outcomes.

Variables	HPV		Kappa Coefficient	*p*
	Yes (n = 64)	No (n = 8)		
CIN1				
yes	22 (34.4)	6 (75.0)	−0.14	0.917
No	42 (65.6)	2 (25.0)		
CIN2				
yes	20 (31.2)	2 (25.0)	0.02	0.413
No	44 (68.8)	6 (75.0)		
CIN3				
yes	20 (31.2)	0 (0.0)	0.09	0.132
No	44 (68.8)	8 (100.0)		

CIN—cervical intraepithelial neoplasia; HPV—human papillomavirus.

**Table 5 cancers-16-03485-t005:** Sensitivity and specificity of ISH II and ISH III in HPV of respective type.

Variable	HPV of Respective Type *				Sensitivity, %	Specificity, %	PPV, %	NPV, %	Accuracy, %
	Yes		No	Total					
ISH II	+	8	64	72	100.00 (63.06–100.00)	0.00 (0.00–5.60)	11.11 (7.72–12.20)	-	11.11 (3.85–18.37)
	−	0	0	0					
ISH III	+	53	19	72	100.00 (93.28–100.00)	0.00 (0.00–17.65)	73.61 (72.03–77.08)	-	73.61 (63.43–83.79)
	−	0	0	0					

PPV—positive predictive value, NPV—negative predictive value. * HPV types 6 or 11 correspond to ISH II and HPV types 16, 18, 31, 33, 35, 45, 52, 56, 58, and 66 correspond to ISH III.

## Data Availability

All source data are available from the corresponding author.

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
