# Peer review of "Evaluation of Automatic Signal Detection of In Situ Hybridization for Detecting HPV DNA in Cervical Tissue Derived from Patients with Cervical Intraepithelial Neoplasia"

_cancers, 2024, doi:10.3390/cancers16203485_

Round 1
Reviewer 1 Report
Comments and Suggestions for Authors
In the present paper, Pruski et al. report on their experience on the automatic signal detection of In Situ Hybridization for detecting HPV DNA in cervical tissue derived from patients with Cervical Intraepithelial Neoplasia. The paper is very interesting, as it provides significant insights on this innovative approach.
Briefly, Authors document that Automatic signal detection of ISH assay cannot be particularly applicable to cervical tissue material, as suggested by diagnostic performance analysis. Authors therefore recommend as a more useful method of confirming the presence of HPV in the cervix the HPV test with genotyping, as it allows for collecting a larger amount of material from the cervical disc and canal. Therefore, "the interpretation of a positive or negative ISH test must be guided in the context of clinical history and morphology".
While conclusions of the Authors can be otherwise acknowledged and eventually shared, the present reviewer is forced to stress that results and discussion sections are affected by significant shortcomings that impair the eventual acceptance of the paper in its current stage of development (according to my point of view, of course). On the other hand, all of the potential issues I've noticed can be addressed in a subsequent revision, and more precisely:
1) Please start results section with more accurate description of the sample (i.e. The sample included data from 72 patients, with a mean age of... etc.)
2) Descriptive description of the sample should be enriched by including data currently included in Table 1
3) Section from row 193 to row 202 is a little bit confusing; I warmly suggest to remove Figure 2, 3 (can be moved to annex material in order to shed some lights on the results, but both are not necessary for understanding the main content) and merge Table 2 and 3, revising the report of the results accordingly;
4) Figure 4 and 5 could be improved (and the corresponding text as well) by including 95%CI of rho, and increasing the size of single dots; please also explain in both labels and text what ISH means as readers could be not particularly familiar with this index.
5) Please revise reporting of data by inverting the cohen's kappa table with Table 5 (as the formula of Cohen's kappa includes data otherwise included in Table 5, the sequence will be more linear)
6) In table 5, you could somehow simplify the reporting omitting negative rows as no "uncertain" results are included
7) Table 6+7 could be merged
8) rows 297 and 298 are particularly important for the content of the paper BUT both should be moved to a "limits" main section of the paper.
9) please revise the flow of discussion section by providing: 4.1 summary of main findings, 4.2 generalizability of results (including text from row 308 onwards), 4.3 limits
Reviewer 2 Report
Comments and Suggestions for Authors
The aim of the submitted manuscript was, as clearly stated in the introduction, to assess the association between HPV infection and an automatic signal detection of ISH for detecting HPV DNA using the In form HPV II and HPV III probe in FFPE cervical tissue specimens with CINs. While the results are not very impressive, no strong correlations have been found between the analyzed parameters, from the technical and methodological points of view the work has been conducted properly. The topic is also within the Cancers scope and the level of this study is acceptable. There are, however, some issues that should be solved before proceeding further.
Lines 5-6, the Authors must decide who is the first and who is the last authors, either Przybylski Marcin or Pruski Dominik.
Lines 375-377, this part must be either updated or removed
Line 169, Table 1, such information are usually provide in the Materials and Methods section
Table 3, why some parts are in yellow (HPV)?
Figures 2 and 3 are not informative and should be either removed or moved to SI.
Lines 237-238, I’d say that this correlation is very weak and should not even be mentioned.
Page 1, the Authors haven’t provided their affiliations
Line 11, “In Situ Hybridization” should be added next to ISH here and not in Line 15, the same applies to HPV. Abbreviations must be explained the first time they appear in the text.
Line 40, this statement should start in a new paragraph
Lines 350-354, those are not the conclusions from the current study but rather some well known facts
Round 2
Reviewer 1 Report
Comments and Suggestions for Authors
Estimated Authors,
I've warmly appreciated the efforts you paid in order to improve your paper in accord to my recommendations.
As a consequence, I'm endorsing its acceptance for full publication.